# Polygenetic risk scores do not add predictive power to clinical models for response to anti-TNFα therapy in inflammatory bowel disease

**Naomi Karmi**[1,2], **Amber Bangma**[1,2], **Lieke M. Spekhorst**[1], **Hendrik M. van Dullemen**[1], **Marijn C. Visschedijk**[1], **Gerard Dijkstra**[1], **Rinse K. Weersma**[1], **Michiel D. Voskuil**[1,2], **Eleonora A. M. Festen**[1,2] *

1 Department of Gastroenterology and Hepatology, University Medical Center Groningen, University of Groningen, Groningen, The Netherlands, 2 Department of Genetics, University Medical Center Groningen, University of Groningen, Groningen, The Netherlands

* e.a.m.festen@umcg.nl

## Abstract

### Background

Anti-tumour necrosis factor alpha (TNFα) therapy is widely used in the management of Crohn's disease (CD) and ulcerative colitis (UC). However, up to a third of patients do not respond to induction therapy and another third of patients lose response over time. To aid patient stratification, polygenetic risk scores have been identified as predictors of response to anti-TNFα therapy. We aimed to replicate the association between polygenetic risk scores and response to anti-TNFα therapy in an independent cohort of patients, to establish its clinical validity.

### Materials and methods

Primary non-response, primary response, durable response and loss of response to anti-TNFα therapy was retrospectively assessed for each patient using stringent definitions. Genome wide genotyping was performed and previously described polygenetic risk scores for primary non-response and durable response were calculated. We compared polygenetic risk scores between patients with primary response and primary non-response, and between patients with durable response and loss of response, using separate analyses for CD and UC.

### Results

Out of 334 patients with CD, 15 (4%) patients met criteria for primary non-response, 221 (66%) for primary response, 115 (34%) for durable response and 35 (10%) for loss of response. Out of 112 patients with UC, 12 (11%) met criteria for primary non-response, 68 (61%) for primary response, 19 (17%) for durable response and 20 (18%) for loss of response. No significant differences in polygenetic risk scores were found between primary non-responders and primary responders, and between durable responders and loss of responders.

**Data Availability Statement:** Raw data is (in part) available at https://ega-archive.org/studies/EGAS00001002702, or upon request.

**Funding:** R.K.W. is supported by a Diagnostics Grant from the Dutch Digestive Foundation (D16-14). www.narcis.nl Yes - played a role in: Conceptualization Data curation Formal analysis Funding acquisition Resources Supervision Validation Writing – review & editing E.A.M.F. is supported by a MLDS Career Development grant (CDG 14-04). www.mlds.nl Yes - played a role in: Conceptualization Data curation Formal analysis Funding acquisition Investigation Methodology Project administration Resources Software Supervision Validation Visualization Writing – original draft Writing – review & editing.

**Competing interests:** I have read the journal's policy and the authors of this manuscript have the following competing interests: G.D. received an unrestricted research grant from Takeda, and received speaker fees from Pfizer and Janssen Pharmaceuticals. R.K.W. acted as consultant for Takeda, received unrestricted research grants from Takeda, Johnson and Johnson, Tramedico and Ferring and received speaker fees from MSD, Abbvie and Janssen Pharmaceuticals. E.A.M.F. received an unrestricted research grant from Takeda. The remaining authors disclose no conflicts. Our competing interests do not alter our adherence to PLOS ONE policies on sharing data and materials.

**Abbreviations:** CD, Crohn's disease; DR, durable response; IBD, inflammatory bowel disease; LOR, loss of response; PNR, primary non-response; PRS, polygenetic risk scores; SNP, single nucleotide polymorphism; TNFα, tumour necrosis factor alpha; UC, ulcerative colitis.

## Conclusions

We could not replicate the previously reported association between polygenetic risk scores and response to anti-TNFα therapy in an independent cohort of patients with CD or UC. Currently, there is insufficient evidence to use polygenetic risk scores to predict response to anti-TNFα therapy in patients with IBD.

## Introduction

Inflammatory bowel disease (IBD), consisting of Crohn's disease (CD) and ulcerative colitis (UC), is a chronic inflammatory disease of the gastrointestinal tract. Although the exact pathogenesis of IBD remains unknown, IBD is thought to be caused by an exaggerated immune response to microbes in the gut in genetically susceptible individuals [1, 2]. Therapy is aimed at inducing and maintaining remission and limiting inflammation-driven damage to the intestinal mucosa.

Management of IBD was revolutionized by the introduction of anti-tumour necrosis factor alpha (anti-TNFα) therapy, initially consisting of infliximab and adalimumab [3]. Unfortunately, response rates of anti-TNFα therapy are low. Up to 30% of all patients do not respond to induction with anti-TNFα therapy (primary non-response [PNR]) and 23–46% of patients lose response over time (loss of response [LOR]) [4]. Furthermore, anti-TNFα therapy can be complicated by adverse events and is a major cost-driver in the management of IBD [5]. Given the low response rates, possible side effects, and the high costs associated with anti-TNFα therapy, there is a need for better prediction of response to anti-TNFα therapy in patients with IBD.

The use of genotype data to predict response to anti-TNFα therapy is increasingly studied, but its clinical utility has not yet been established [6]. Two previous studies have identified sets of genetic variants associated with response to anti-TNFα therapy in patients with CD or UC. These genetic variants were incorporated into separate polygenetic risk scores (PRS) for CD and UC. A model combining clinical data and PRS showed a more accurate prediction of PNR and durable response (DR) to anti-TNFα therapy than a clinical-only model [7, 8]. To validate these PRS, the current study aims to replicate the association of PRS with response to anti-TNFα therapy in an independent cohort of patients with CD and UC.

## Materials and methods

### Study population

Patients were included as part of the 1000IBD cohort. The 1000IBD cohort is a prospective cohort of patients with IBD undergoing treatment in the University Medical Center Groningen, the Netherlands [9]. Patients with CD or UC treated with anti-TNFα therapy (infliximab and/or adalimumab) between October 1999 and May 2020 were included in this study. Each patient had been diagnosed with CD or UC by their gastroenterologist using endoscopic, histological, or radiological data, or a combination of these. Written consent was obtained from all patients, and the study was approved by the medical ethical board of the University Medical Center Groningen (PSI-UMCG [IRB no 08/279]).

## Case criteria

Each patient was classified as either case or control for both PNR and DR. Response was based upon the physician global assessment using a combination of clinical, radiologic, endoscopic and laboratory data. Response criteria for definite, probable and possible cases were defined in line with previous studies [7, 8] (S1 File). Patients received anti-TNFα therapy with standard induction dosing of infliximab at week 0, 2, and 6 and for adalimumab at week 0 and 2.

Definite PNR was defined as non-response after a period of up to 16 weeks after starting anti-TNFα therapy accompanied by an alteration of therapeutic approach (addition or escalation of corticosteroids, switch to a different agent or surgery). Possible PNR cases required matching of definite case criteria but allowed continuation of anti-TNFα therapy after 16 weeks, despite no clear signs of response. Patients with a primary response, which was evaluated after a minimum of three infusions between 12 and 16 weeks after start of treatment, were included as controls for PNR. Only definite PNR and controls, from here on referred to as primary responders, were included in subsequent analyses.

Definite DR was defined as maintenance of response to anti-TNFα therapy until latest follow-up, for at least 24 months after initiation of anti-TNFα therapy. Probable DR cases were defined as maintenance of response to anti-TNFα therapy for at least 24 months after initiation, but patients were included if available data suggested LOR after the 24-month time point. Patients who ceased treatment prior to the 24-month time point due to LOR or due to adverse events related to LOR (such as immunogenicity) were included as controls for DR. Patients that ceased treatment prior to the 24-month time point due to adverse events unrelated to LOR (such as non-IBD related infections) were excluded from analyses to assess DR. Only definite DR and controls, from here on referred to as patients with LOR, were included in subsequent analyses.

## Data collection

Information was collected on age, age at diagnosis, sex, duration of disease at initiation of anti-TNFα therapy, type of anti-TNFα therapy, and concomitant use of an immunomodulator (azathioprine, 6-mercaptopurine or methotrexate). All patients were genotyped using the Infinium Global Screening Array (Illumina, San Diego, CA, USA). After extensive quality control (S2 File) and pre-phasing with the Eagle2 algorithm, genotype data were imputed to the Haplotype Reference Consortium reference panel using the Michigan Imputation server [10]. After post-imputation quality control, 12,130,010 genetic variants with a minor allele frequency > 0.1% remained. To limit bias from population stratification, only patients clustering with genetic data from non-Finnish European individuals were included, using the 1KG European dataset as the external reference panel [11].

## Statistical analysis

Genetic variants incorporated in previously published PRS [7, 8] were selected from the imputed genetic data based on landmark IBD genotype-phenotype studies that identified IBD susceptibility loci [12–14]. In total, we identified all out of 50 previously described variants (S1–S4 Tables). Separate analyses were performed for CD and UC, and for PNR and DR, creating four distinct analyses. Separate weighted polygenetic risk scores for PNR and DR were calculated as the cumulative sum of the product of the log-odds ratio and allele burden for each of the risk genetic variants, also called single nucleotide polymorphisms (SNPs). Clinical characteristics were presented as means or medians and standard deviations or interquartile ranges for continuous variables, and as numbers and percentages for categorical variables. Cases and controls for PNR and DR were first compared using univariate analysis. For continuous

variables, an unpaired t-test for normally distributed variables and a Wilcoxon's Rank Sum test for non-normally distributed variables was used. For categorical variables, a chi-square test was used. Significant variables (P < 0.05) in univariate analyses were included in subsequent multivariate analyses. All statistical analyses were performed using R 3.5.1 (R Foundations for Statistical Computing, Vienna, Austria). We performed a power calculation by selecting SNPs within a range of p-values from the reference CD study and calculated how much power we had to detect these SNPs in our present study.

## Results

### Crohn's disease

**Predictors of primary non-response.** We identified 334 patients with CD who had received anti-TNFα therapy. Of these patients, we identified 15 (4%) patients with PNR and 221 (66%) as primary responders. Remaining patients did not meet the stringent case-control criteria. No significant differences between patients with PNR and primary responders in age, age at diagnosis, disease duration, sex, type of anti-TNFα therapy and concomitant therapy were found upon univariate analyses (Table 1). Furthermore, PRS were similar between patients with PNR and primary responders (0.77 [1.2] vs 1.21 [1.9]; P = 0.1955) (Fig 1A).

**Predictors of durable response.** Of the 334 patients with CD who had received anti-TNFα therapy, 115 (34%) were classified as patients with DR and 35 (10%) as patients with LOR. Remaining patients did not meet stringent case-control criteria. There were no significant differences between patients with DR and patients with LOR in age, age at diagnosis, disease duration, sex and type of anti-TNFα therapy (Table 2). The use of concomitant therapy was independently predictive of DR in patients with CD (OR 3.01 [95% CI 0.40–2.74]; P = 0.0062). PRS were similar between patients with DR and patients with LOR (0.58 [1.9] vs 0.76 [1.7]; P = 0.4666) (Fig 1B).

**Table 1. Comparison of characteristics of patients with Crohn's disease exposed to anti-tumour necrosis factor alpha therapy with primary non-response and primary response.**

| | Primary non-responders (n = 15) | Primary responders (n = 221) | P-value |
|---|---|---|---|
| Age, median (IQR) (in years) | 45 (11) | 44 (14) | 0.7873 |
| Age at diagnosis, median (IQR) (in years) | 25 (12) | 24 (13) | 0.5602 |
| Disease duration, median (IQR) (in years) | 3 (6) | 4 (9) | 0.3394 |
| Sex | | | 0.1618 |
| • Male, No. (%) | 3 (20) | 84 (38) | |
| • Female, No. (%) | 12 (80) | 137 (62) | |
| First anti-TNFα therapy, No. (%) | | | 0.3815 |
| • Adalimumab | 2 (20) | 51 (23) | |
| • Infliximab | 13 (80) | 170 (77) | |
| Combination immunosuppression (azathioprine, 6-mercaptopurine, methotrexate), No. (%) | 7 (47) | 123 (56) | 0.4982 |
| Weighted PRS for PNR in CD, mean (SD) | 0.77 (1.2) | 1.21 (1.9) | 0.1955 |

Abbreviations: IQR, inter-quartile range; No., number; anti-TNFα, anti-tumour necrosis factor alpha; PRS, polygenetic risk score; PNR, primary non-response; CD, Crohn's disease; SD, standard deviation.

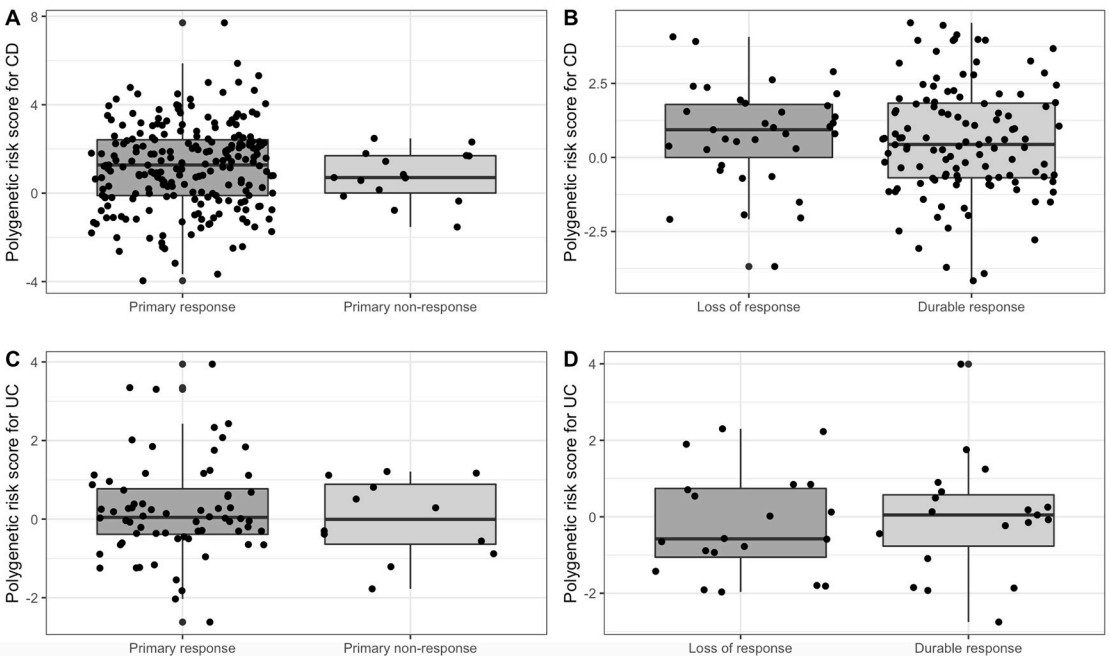

**Fig 1. Boxplot representing the distribution of polygenetic risk scores for response in patients with inflammatory bowel disease.** A Boxplot representing the distribution of polygenetic risk score for primary non-response in patients with Crohn's disease with primary non-response and primary response; B Boxplot representing the distribution of polygenetic risk score for durable response in patients with Crohn's disease with durable response and loss of response; C Boxplot representing the distribution of polygenetic risk score for primary non-response in patients with ulcerative colitis with primary non-response and primary response; D Boxplot representing the distribution of polygenetic risk score for durable response in patients with ulcerative colitis with durable response and loss of response.

## Ulcerative colitis

**Predictors of primary non-response.** We identified 112 patients with UC who had received anti-TNFα therapy. Of these patients, we identified 12 (11%) patients with PNR and

**Table 2. Comparison of characteristics of patients with Crohn's disease exposed to anti-tumour necrosis factor alpha therapy with durable response and loss of response.**

|  | Durable response (n = 115) | Loss of responders (n = 35) | P-value |
|---|---|---|---|
| Age, median (IQR) (in years) | 41 (20) | 45 (17) | 0.5648 |
| Age at diagnosis, median (IQR) (in years) | 24 (12) | 25 (11) | 0.6397 |
| Disease duration, median (IQR) (in years) | 3 (9) | 4 (15) | 0.4781 |
| Sex |  |  | 0.1894 |
| • Male, No. (%) | 47 (41) | 10 (29) |  |
| • Female, No. (%) | 68 (59) | 25 (71) |  |
| First anti-TNFα therapy, No. (%) |  |  | 0.2750 |
| • Adalimumab | 20 (17) | 9 (26) |  |
| • Infliximab | 95 (83) | 26 (74) |  |
| Combination immunosuppression (azathioprine, 6-mercaptopurine, methotrexate), No. (%) | 73 (63) | 13 (37) | 0.0058 |
| Weighted PRS for DR in CD, mean (SD) | 0.58 (1.9) | 0.76 (1.7) | 0.4666[a] |

Abbreviations: IQR, inter-quartile range; No., number; anti-TNFα, anti-tumour necrosis factor alpha; PRS, polygenetic risk score; DR, durable response; CD, Crohn's disease; SD, standard deviation.

[a]P-value from multivariate analysis including the use of a concomitant immunomodulator.

**Table 3. Comparison of characteristics of patients with ulcerative colitis exposed to anti-tumour necrosis factor alpha therapy with primary non-response and primary response.**

| | Primary non-responders (n = 12) | Primary responders (n = 68) | P-value |
|---|---|---|---|
| Age, mean (SD) (in years) | 44 (12) | 49 (15) | 0.3717 |
| Age at diagnosis, median (IQR) (in years) | 29 (23) | 30 (24) | 0.8510 |
| Disease duration, median (IQR) (in years) | 3 (5) | 5 (9) | 0.0711 |
| Sex | | | 0.7298 |
| • Male, No. (%) | 5 (42) | 32 (47) | |
| • Female, No. (%) | 7 (58) | 36 (53) | |
| First anti-TNFα therapy, No. (%) | | | 0.8410 |
| • Adalimumab | 2 (17) | 13 (19) | |
| • Infliximab | 10 (83) | 55 (81) | |
| Combination immunosuppression (azathioprine, 6-mercaptopurine, methotrexate), No. (%) | 5 (42) | 30 (44) | 0.8746 |
| Weighted PRS for PNR in UC, median (IQR) | -0.01 (1.5) | 0.03 (1.2) | 0.7464 |

Abbreviations: SD, standard deviation; IQR, inter-quartile range; No., number; anti-TNFα, anti-tumour necrosis factor alpha; PRS, polygenetic risk score; PNR, primary non-response; UC, ulcerative colitis.

68 (61%) as primary responders. Remaining patients did not meet stringent case-control criteria. There were no significant differences between non-responders and primary responders in age, age at diagnosis, disease duration, sex, type of anti-TNFα therapy and concomitant therapy (Table 3). PRS were similar between patients with PNR and primary responders (-0.01 [IQR 1.5] vs 0.03 [IQR 1.2], $P = 0.7464$) (Fig 1C).

**Predictors of durable response.** Of the 112 patients with UC that received anti-TNFα therapy, 19 (17%) were classified as patients with DR and 20 (18%) as patients with LOR. Remaining patients did not meet stringent case-control criteria. There were no significant differences between patients with DR and patients with LOR in age at diagnosis, disease duration, sex and type of anti-TNFα therapy (Table 4). A lower age and the use of concomitant immunosuppressive therapy were predictive of DR in univariate analyses ($P = 0.0407$ and $P = 0.0379$, respectively), but these associations were lost in multivariate analyses ($P = 0.1780$ and $P = 0.1710$, respectively). PRS were similar in patients with DR and patients with LOR (-0.04 [1.5] vs -0.19 [1.4]; $P = 0.8840$) (Fig 1D).

## Discussion

This present study aimed to replicate previously described PRS for PNR and DR to anti-TNFα therapy in patients with IBD. Using an independent cohort of patients with CD and UC, we could not replicate the association of PRS with response to anti-TNFα therapy. Our findings show that there is currently insufficient scientific basis for the use of PRS as an addition to clinical models to predict response to anti-TNFα therapy in patients with IBD.

To compare our data to previously published PRS, we used similar methodology to generate the PRS. However, our methodology had stricter definitions of cases and weighted PRS were calculated for CD and UC. Therefore, the methodology used may explain why we could not identify any predictive power of PRS with regard to response to anti-TNFα therapy. PRS in the original studies were based on genetic variants associated with PNR and DR in patients with CD and UC. Known IBD risk genetic variants were incorporated if they were associated with PNR or DR with $P < 0.05$. Additionally, non-IBD risk genetic variants were selected from Immunochip loci if they were associated with PNR or DR with a $P < 1 \times 10^{-4}$ for CD and $P < 1 \times 10^{-6}$ for UC [7, 8]. For UC, genetic variants were weighted by the sum of log-odds

**Table 4. Comparison of characteristics of patients with ulcerative colitis exposed to anti-tumour necrosis factor alpha therapy with durable response and loss of response.**

| | Durable response (n = 19) | Loss of responders (n = 20) | P-value |
|---|---|---|---|
| Age, mean (SD) (in years) | 44 (15) | 54 (16) | 0.0407 |
| Age at diagnosis, mean (SD) (in years) | 27 (16) | 34 (16) | 0.2654 |
| Disease duration, median (IQR) (in years) | 5 (6) | 11 (14) | 0.2087 |
| Sex | | | 0.1481 |
| • Male, No. (%) | 7 (37) | 12 (60) | |
| • Female, No. (%) | 12 (63) | 8 (40) | |
| First anti-TNFα therapy, No. (%) | | | 0.2193 |
| • Adalimumab | 6 (32) | 3 (15) | |
| • Infliximab | 13 (68) | 17 (85) | |
| Combination immunosuppression (azathioprine, 6-mercaptopurine, methotrexate), No. (%) | 12 (63) | 6 (30) | 0.0379 |
| Weighted PRS for DR in UC, mean (SD) | -0.04 (1.5) | -0.19 (1.4) | 0.8840[a] |

Abbreviations: SD, standard deviation; IQR, inter-quartile range; No., number; anti-TNFα, anti-tumour necrosis factor alpha; PRS, polygenetic risk score; DR, durable response; UC, ulcerative colitis.

[a]P-value from multivariate analysis including age and the use of a concomitant immunomodulator.

ratio and allele burden to create PRS, whereas for CD, selected genetic variants were combined into unweighted PRS. Since genetic variants may have different effect sizes, the use of a weighted PRS is preferred over non-weighted PRS. Therefore, in this present study, genetic variants in both risk scores for CD and UC were weighted by the sum of known log-odds ratio and allele burden [7, 8]. The fact that both our scores were weighted, could add to the differences found in response associations compared to the prior studies. Furthermore, we believe that future studies exploring PRS to predict response to anti-TNFα therapy could benefit from much stricter p-value thresholds. Using these stricter p-value thresholds, larger and independent cohorts of patients should limit false-positive findings. A combined predictive model including PRS and clinical data has been previously associated with response to anti-TNFα therapy [7, 8]. These prior studies have focused on creating PRS to predict response to anti-TNFα therapy for CD and UC separately, using IBD specific and non-specific risk alleles. However, one could wonder why almost no overlap in genetic variants was observed between the scores for CD and UC, while the mechanism of response to anti-TNFα are expected to be similar in both IBD phenotypes. In addition, thresholds for significance used in the discussed studies were relatively lenient (e.g. P < 0.05). This lack of overlap suggests that the selected genetic variants for the PRS may be false positives rather than a reflection of true biologic signals. Future studies should focus on creating international collaborations to achieve larger sample sizes, enabling the use of more stringent significance thresholds and reproducibility.

This study emphasizes the importance of replication when studying clinically relevant scientific findings. Before a genetic test can be implemented into clinical care, its clinical validity has to be established. Key in this is replication of the genetic association in an independent cohort [3]. In recent years, some replications of earlier studies have been unable to reproduce important scientific findings, leading to a "reproducibility crisis" [15]. This emphasizes the importance of not only publishing the original positive results, but also those that fail in replication [16, 17].

There are several strengths to this study. First, our cohort provides detailed phenotypic data in combination of genome-wide genetic data. Second, together with the detailed description of the cohort, long-term follow-up enabled stringent definitions for PNR and DR. Third, with

the aim to replicate previous findings, we used similar definitions for PNR, and DR as used in the reference studies [7, 8].

However, our study has several limitations which should be taken into account when interpreting the results. The retrospective character of this study posed a challenge in case-control adjudication. Ideally, patients are prospectively followed to allow accurate testing of case-control definitions. Immunogenicity or low trough levels may cause therapeutic failure and would preferably be ruled out prior to adjudicating patients as true primary non-responder [18]. Our cohort lacks detailed data on drug trough levels and anti-drug antibodies, which could add bias to our results. Lastly, the relatively small sample size of this present study might be contributing to the lack of significance presented here. Nevertheless, we identified all out of 50 previously described variants (i.e. genotypes present in dataset with expected minor allele frequencies). This identification implicates that our sample size was large enough to comment on the association between PRS and response to anti-TNFα therapy, as identified in the reference studies [7, 8]. Furthermore, a post-hoc power calculation on the prior studies PRS showed >80% power to detect more than half of the genetic variants in the present CD study.

In conclusion, we were not able to replicate the association of PRS with response to anti-TNFα therapy for patients with IBD, demonstrating insufficient scientific evidence for the use of genetic data to successfully predict response to anti-TNFα therapy in IBD. Low response rates and high costs remain a challenge for anti-TNFα therapy, which emphasizes the need for better prediction tools of response [5, 19]. Future research based on international collaborations should focus on providing better understanding and prediction of therapies for IBD.

## Supporting information

**S1 File. Case and control criteria.**
(DOCX)

**S2 File. Genetic data generation.**
(DOCX)

**S3 File.**
(DOCX)

**S1 Table. Single-nucleotide polymorphisms associated with primary non-response in patients with Crohn's disease.** SNPs were selected in a prior study at p-value < 0.05 among 163 IBD risk alleles and p-value of $<1 \times 10{-}4$ among the immunochip. For the weighted analysis of PRS we used the previously calculated odds ratios [1]. a = our study in CD. b = the prior study in CD. Abbreviations: SNP, single-nucleotide polymorphism; Freq., Frequency; PNR, primary non-response; PR, primary response; IBD, inflammatory bowel disease; CD, Crohn's disease.
(DOCX)

**S2 Table. Single-nucleotide polymorphisms associated with durable response in patients with Crohn's disease.** SNPs were selected in a prior study at p-value < 0.05 among 163 IBD risk alleles and p-value of $<1 \times 10{-}4$ among the immunochip. For the weighted analysis of PRS we used the previously calculated odds ratios [1]. a = our study in CD. b = the prior study in CD. Abbreviations: SNP, single-nucleotide polymorphism; Freq., Frequency; DR, durable response; LOR, loss of response; IBD, inflammatory bowel disease; CD, Crohn's disease.
(DOCX)

**S3 Table. Single-nucleotide polymorphisms associated with primary non-response in patients with ulcerative colitis.** SNPs were selected in a prior study at p-value < 0.05 among

201 IBD risk alleles and p-value of $<1 \times 10-6$ among the immunochip. For the weighted analysis of PRS we used the previously calculated odds ratios [2]. a = our study in UC. b = the prior study in UC. Abbreviations: SNP, single-nucleotide polymorphism; Freq. Frequency; PNR, primary non-response; PR, primary response; IBD, inflammatory bowel disease; UC, ulcerative colitis.
(DOCX)

**S4 Table. Single-nucleotide polymorphisms associated with durable response in patients with ulcerative colitis.** SNPs were selected in a prior study at p-value < 0.05 among 201 IBD risk alleles and p-value of $<1 \times 10-6$ among the immunochip. For the weighted analysis of PRS we used the previously calculated odds ratios [2]. a = our study in UC. b = the prior study in UC. Abbreviations: SNP, single-nucleotide polymorphism; Freq. Frequency; DR, durable response; LOR, loss of response; IBD, inflammatory bowel disease; UC, ulcerative colitis.
(DOCX)

## Acknowledgments

The authors thank all participants of the 1000IBD cohort.

## Author Contributions

**Conceptualization:** Naomi Karmi, Amber Bangma, Lieke M. Spekhorst, Hendrik M. van Dullemen, Marijn C. Visschedijk, Gerard Dijkstra, Rinse K. Weersma, Michiel D. Voskuil, Eleonora A. M. Festen.

**Data curation:** Naomi Karmi, Amber Bangma, Lieke M. Spekhorst, Hendrik M. van Dullemen, Marijn C. Visschedijk, Gerard Dijkstra, Rinse K. Weersma, Michiel D. Voskuil, Eleonora A. M. Festen.

**Formal analysis:** Naomi Karmi, Amber Bangma, Lieke M. Spekhorst, Hendrik M. van Dullemen, Marijn C. Visschedijk, Gerard Dijkstra, Rinse K. Weersma, Michiel D. Voskuil, Eleonora A. M. Festen.

**Funding acquisition:** Naomi Karmi, Amber Bangma, Lieke M. Spekhorst, Hendrik M. van Dullemen, Marijn C. Visschedijk, Gerard Dijkstra, Rinse K. Weersma, Michiel D. Voskuil, Eleonora A. M. Festen.

**Investigation:** Naomi Karmi, Amber Bangma, Michiel D. Voskuil, Eleonora A. M. Festen.

**Methodology:** Naomi Karmi, Amber Bangma, Lieke M. Spekhorst, Hendrik M. van Dullemen, Marijn C. Visschedijk, Gerard Dijkstra, Michiel D. Voskuil, Eleonora A. M. Festen.

**Project administration:** Naomi Karmi, Amber Bangma, Michiel D. Voskuil, Eleonora A. M. Festen.

**Resources:** Naomi Karmi, Amber Bangma, Lieke M. Spekhorst, Rinse K. Weersma, Michiel D. Voskuil, Eleonora A. M. Festen.

**Software:** Naomi Karmi, Amber Bangma, Michiel D. Voskuil, Eleonora A. M. Festen.

**Supervision:** Naomi Karmi, Amber Bangma, Hendrik M. van Dullemen, Rinse K. Weersma, Michiel D. Voskuil, Eleonora A. M. Festen.

**Validation:** Naomi Karmi, Amber Bangma, Lieke M. Spekhorst, Hendrik M. van Dullemen, Marijn C. Visschedijk, Gerard Dijkstra, Rinse K. Weersma, Michiel D. Voskuil, Eleonora A. M. Festen.

**Visualization:** Naomi Karmi, Amber Bangma, Michiel D. Voskuil, Eleonora A. M. Festen.

**Writing – original draft:** Naomi Karmi, Amber Bangma, Michiel D. Voskuil, Eleonora A. M. Festen.

**Writing – review & editing:** Naomi Karmi, Amber Bangma, Lieke M. Spekhorst, Hendrik M. van Dullemen, Marijn C. Visschedijk, Gerard Dijkstra, Rinse K. Weersma, Michiel D. Voskuil, Eleonora A. M. Festen.

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
