## [Decision Letter · Decision Letter 0]

2 Mar 2021

PONE-D-20-37588

Insufficient evidence that polygenetic risk scores can be used to predict response to anti-TNFα therapy in inflammatory bowel disease

PLOS ONE

Dear Dr. Festen,

Thank you for submitting your manuscript to PLOS ONE. After careful consideration, we feel that it has merit but does not fully meet PLOS ONE’s publication criteria as it currently stands. Therefore, we invite you to submit a revised version of the manuscript that addresses the points raised during the review process.

Considering the Reviewer criticisms, the Authors should focus their attention to reply to major comments and to clarify the power of their study. A negative finding is a result, but it needs to be  statistically  verified.

We look forward to receiving your revised manuscript.

Kind regards,

Cinzia Ciccacci

Academic Editor

PLOS ONE

"I have read the journal's policy and the authors of this manuscript have the following competing interests:

G.D. received an unrestricted research grant from Takeda, and received speaker fees from Pfizer and Janssen Pharmaceuticals.

R.K.W. acted as consultant for Takeda, received unrestricted research grants from Takeda, Johnson and Johnson, Tramedico and Ferring and received speaker fees from MSD, Abbvie and Janssen Pharmaceuticals. E.A.M.F. received an unrestricted research grant from Takeda.

The remaining authors disclose no conflicts. "

Reviewers' comments:

Reviewer's Responses to Questions

**Comments to the Author**

1. Is the manuscript technically sound, and do the data support the conclusions?

Reviewer #1: Yes

2. Has the statistical analysis been performed appropriately and rigorously? 

Reviewer #1: No

3. Have the authors made all data underlying the findings in their manuscript fully available?

Reviewer #1: Yes

4. Is the manuscript presented in an intelligible fashion and written in standard English?

Reviewer #1: Yes

5. Review Comments to the Author

Reviewer #1: In their manuscript “Insufficient evidence that polygenetic risk scores can be used to predict response to anti-TNFα therapy in inflammatory bowel disease”, the authors aim to replicate previous findings on the predictive value of IBD genetic susceptibility loci for therapy response. In contrast to the previous work, their results demonstrate no such predictive capability, which is a (negative) finding well worth publishing to enable a balanced picture in this research area. However, their study designed has several limitations, as pointed out below, which need to be addressed and properly discussed in the manuscript.

Minor concerns:

• Line 59: the authors state high toxicity of anti-TNF therapy; however, the side effects of anti-TNF are considerably lower, relative to conventional immunotherapy with agents such as thiopurines – this has to be clarified in the text

• Throughout, the authors should state the allele frequencies of risk SNPs in their study populations

• The definition of response is not clear, as the authors state it was due to the individual clinician’s perspective – the methods will need to state precise and uniform criteria for response, otherwise it will be hard to rule out bias through non-uniform response definition

• Whenever data is presented as box plots, all data points should be shown

• Given the topic of the manuscript, the authors will need to reference all landmark studies that identified IBD susceptibility loci (e.g. Jostins et al, to only name one)

Major concerns:

• Line 64f: the authors reference two studies demonstrating a predictive value of “PRS” for therapy response – however in both of these studies, clinical parameters were incorporated into the predictive model, in addition to genetics; this has to be clarified, as the title of the manuscript suggests the use of genetic markers only

• Definition of response does not consider the anti-TNF drug levels – patients should only be considered true non-responders, if they had adequate trough drug levels but still did not respond; otherwise a therapy failure due to pharmacological aspects rather than genetics cannot be ruled out

• The # of patients is not particularly high, so probably study was underpowered to detect many of the other IBD risk SNPs; this has to be stated and discussed in the manuscript; in particular it will require altering the author’s conclusion that genetic risk loci are not predictive of therapy response in IBD – they can only comment on the ones they reliably detected

• Related to the point above, a power calculation should be presented and how this impacts on the authors’ conclusions

• To increase power, the author’s should conduct a meta-analysis (including data from Refs 7 and 8)

6. PLOS authors have the option to publish the peer review history of their article (what does this mean?). If published, this will include your full peer review and any attached files.

Reviewer #1: No

---

## [Author Response · Author response to Decision Letter 0]

25 May 2021

We would like to express our thanks for the reviewer’ efforts and invested time to review our manuscript. The mentions concerns were useful and really added to our manuscript. All together it resulted in a much-improved version of the manuscript. 

Reviewer #1: 

In their manuscript “Insufficient evidence that polygenetic risk scores can be used to predict response to anti-TNFα therapy in inflammatory bowel disease”, the authors aim to replicate previous findings on the predictive value of IBD genetic susceptibility loci for therapy response. In contrast to the previous work, their results demonstrate no such predictive capability, which is a (negative) finding well worth publishing to enable a balanced picture in this research area. However, their study designed has several limitations, as pointed out below, which need to be addressed and properly discussed in the manuscript.

Minor concerns:

Q1 Line 59: the authors state high toxicity of anti-TNF therapy; however, the side effects of anti-TNF are considerably lower, relative to conventional immunotherapy with agents such as thiopurines – this has to be clarified in the text

R1 We would like to thank the reviewer for bringing this inaccurate adjective to our attention and for allowing us to convey nuance to the text. We agree that anti-TNF� therapy is generally well-tolerated in clinical practice, although it increases the susceptibility to severe infections, possibly melanoma skin cancer, and treatment-related complications, such as lupus-like syndromes and allergic reactions. To write a truer definition of adverse events in anti-TNF� therapy we replaced ‘high toxicity’ with ‘possible side effects’ at lines 59-60. 

Q2 Throughout, the authors should state the allele frequencies of risk SNPs in their study populations.

R2 Thank you for this suggestion. All allele frequencies are now available in our supplemental tables. Furthermore, we adjusted the table legends of tables S1-S4 in our supporting information to clarify the source of the frequencies, p-values and odds ratios. 

Q3 The definition of response is not clear, as the authors state it was due to the individual clinician’s perspective – the methods will need to state precise and uniform criteria for response, otherwise it will be hard to rule out bias through non-uniform response definition

R3 We agree with this reviewer that the definition used to assess response is crucial for genotype-phenotype studies like the present study. For our retrospective study, we have used the 1000IBD cohort, which we described previously. In this cohort, clinical, radiological, endoscopic, laboratory data, as well as the clinician’s perspective, are prospectively collected in a uniform format. Using a combination of these data, we defined stringent definitions of response, which we present in the supporting information (S1 File. Case and control criteria). 

We agree that defining response is preferably done prospectively and based on validated scores, calprotectin, CRP and/or endoscopy. The aim of this study, however, was to replicate previously identified genotype-phenotype interactions. These previous studies are also retrospective in nature, using physician’s Global Assessment as a measure of response. Therefore, we defined response in a similar fashion to these previous studies. We discuss this issue of response definition in the discussion section of our manuscript at lines 290-292. 

Q4 Whenever data is presented as box plots, all data points should be shown

R4 We agree with the reviewer that Figure 1, “Boxplot representing a distribution of polygenetic risk scores for response in patients with inflammatory bowel disease”, should be altered. We have added all data points to the box plots. 

Q5 Given the topic of the manuscript, the authors will need to reference all landmark studies that identified IBD susceptibility loci (e.g. Jostins et al, to only name one)

R5 We thank the reviewer for this suggestion, and have added the following landmark IBD genotype-phenotype studies to our manuscript (lines 121-123):

- de Lange KM, Moutsianas L, Lee JC, et al. Genome-wide association study implicates immune activation of multiple integrin genes in inflammatory bowel disease. Nat Genet. 2017;49(2):256‐261.

- Liu JZ, van Sommeren S, Huang H, et al; International IBD Genetics Consortium (IIBDGC). Association analyses identify 38 susceptibility loci for inflammatory bowel disease and highlight shared genetic risk across populations. Nat Genet. 2015;47:979–86.

- Jostins L, Ripke S, Weersma RK, et al; International IBD Genetics Consortium (IIBDGC). Host-microbe interactions have shaped the genetic architecture of inflammatory bowel disease. Nature. 2012;491:119–24.

Major concerns:

Q6 Line 64f: the authors reference two studies demonstrating a predictive value of “PRS” for therapy response – however in both of these studies, clinical parameters were incorporated into the predictive model, in addition to genetics; this has to be clarified, as the title of the manuscript suggests the use of genetic markers only

R6 We acknowledge that both prior studies incorporated genetics as an addition to clinical covariates into a predictive model of response to anti-TNF� therapy in patients with IBD. Both prior studies compared the performance of a combined clinical-genetic model with clinical-only and genetic-only models. We now discuss this in our introduction (lines 65-68) and discussion (lines 266-268). In our present study, we used univariate analyses to identify clinical parameters predictive of response. Our area under the receiver operating characteristics (AUROC) curve models demonstrate that a combined clinical-genetic model predicts durable response in patients with CD better than a genetic-only model (AUROC 0.66 vs. 0.55, P= 0.0479). However, the incorporated PRS of these models was non-significant in univariate analysis. Moreover, in patients with UC a combined clinical-genetic model did not predict durable response better than a genetic-only model. 

With our aim to replicate previous findings, we initially formulated the title of our manuscript in a similar fashion to the studies referenced. However, we agree with this reviewer that this formulation may not be accurate. We have changed the title of our manuscript to ‘Polygenetic risk scores do not add predictive power to clinical models for response to anti-TNF� therapy in inflammatory bowel disease”.

Q7 Definition of response does not consider the anti-TNF drug levels – patients should only be considered true non-responders, if they had adequate trough drug levels but still did not respond; otherwise a therapy failure due to pharmacological aspects rather than genetics cannot be ruled out

R7 We agree with the reviewer that multiple aspects may contribute to therapy failure. Indeed, immunogenicity or low trough levels may cause therapeutic failure in induction remission of severe disease or fistula-disease and then should be ruled out prior to adjudicating patients as ‘true non-responders’. Unfortunately, data regarding trough levels and anti-drug antibodies are only common practice since 2017. Therefore, data on trough levels and anti-drug antibodies are only available for a small proportion of our cohort. To increase patient numbers, and thereby power, we included all patients from our cohort that matched our inclusion criteria, despite the lack of data on trough levels or anti-drug antibodies. In fact, the CD reference study we aimed to validate also included patients without these data. Moreover, the UC reference study showed no associations between response and infliximab trough level in a small subgroup. However, we agree that genetic variants may contribute to therapeutic failure due to immunogenicity (Sazonovs A. et al, 2020). These signals may remain undetected in our present study, which could add bias to our results. We discuss this in our revised discussion at lines 292-295. 

Q8 The # of patients is not particularly high, so probably study was underpowered to detect many of the other IBD risk SNPs; this has to be stated and discussed in the manuscript; in particular it will require altering the author’s conclusion that genetic risk loci are not predictive of therapy response in IBD – they can only comment on the ones they reliably detected

R8 We agree with this reviewer’s observation that our study is limited by its relatively small size. Indeed, using a cohort of limited sample size limits our power to detect genotype-phenotype associations of genetic variants with lower allele frequencies or lower effect sizes. 

However, adding a power calculation, as suggested in R9, will show the level of power we have to detect at least half of the previously selected SNPs. This study aimed to validate previously described associations between polygenetic risk scores and response to anti-TNF� therapy. We identified all out of 50 previously described variants and incorporated these into separate PRS for CD and UC. This implicates that our sample size was large enough to comment on the association between PRS and response to anti-TNF� therapy, as identified in the reference studies. However, we agree that genotype-phenotype associations of genetic variants outside this selection may remain undetected. 

While UC and CD have a relatively shared genetic background, it is surprising that the genetic variants in previously identified PRS for CD and UC, respectively, do not overlap. This lack of overlap suggests that the selected genetic variants for the PRS may be false positives rather than a reflection of true biologic signals. We discuss this in our revised manuscript on lines 273-275 and 296-300. 

Q9 Related to the point above, a power calculation should be presented and how this impacts on the authors’ conclusions

R9 To further assess the power of our study we added a power calculation as suggested by the reviewer. We selected different variants in the range of p-values from the reference CD study and calculated how much power we had to detect these variants in our present study (GAS Power Calculator - Skol et. al, 2006). Unfortunately, despite our request to the authors, we did not have access to the allele frequencies of genetic variants used in the PRS of the UC reference study. Without these data, we could not reliably perform a power calculation for our present UC study. 

Below is a visual presentation of our power to detect the prior selected genetic variants in CD. Our power ranged from 93% to 100%, and for more than half of the genetic variants we had >80% power to detect them. 

Q10 To increase power, the author’s should conduct a meta-analysis (including data from Refs 7 and 8)

R10 Thank you for this suggestion. We agree that a proper meta-analyse would be of great value. Unfortunately, after several requests, we could not obtain the relevant data from the reference studies.

---

## [Decision Letter · Decision Letter 1]

18 Aug 2021

Polygenetic risk scores do not add predictive power to clinical models for response to anti-TNF a  therapy in inflammatory bowel disease

PONE-D-20-37588R1

Dear Dr. Festen,

We’re pleased to inform you that your manuscript has been judged scientifically suitable for publication and will be formally accepted for publication once it meets all outstanding technical requirements.

Kind regards,

Cinzia Ciccacci

Academic Editor

PLOS ONE

Additional Editor Comments (optional):

Reviewers' comments:

Reviewer's Responses to Questions

**Comments to the Author**

1. If the authors have adequately addressed your comments raised in a previous round of review and you feel that this manuscript is now acceptable for publication, you may indicate that here to bypass the “Comments to the Author” section, enter your conflict of interest statement in the “Confidential to Editor” section, and submit your "Accept" recommendation.

Reviewer #1: All comments have been addressed

2. Is the manuscript technically sound, and do the data support the conclusions?

Reviewer #1: Yes

3. Has the statistical analysis been performed appropriately and rigorously? 

Reviewer #1: Yes

4. Have the authors made all data underlying the findings in their manuscript fully available?

Reviewer #1: Yes

5. Is the manuscript presented in an intelligible fashion and written in standard English?

Reviewer #1: Yes

6. Review Comments to the Author

Reviewer #1: (No Response)

7. PLOS authors have the option to publish the peer review history of their article (what does this mean?). If published, this will include your full peer review and any attached files.

Reviewer #1: No

---

## [Editor Report · Acceptance letter]

25 Aug 2021

PONE-D-20-37588R1 

Polygenetic risk scores do not add predictive power to clinical models for response to anti-TNFa therapy in inflammatory bowel disease 

Dear Dr. Festen:

I'm pleased to inform you that your manuscript has been deemed suitable for publication in PLOS ONE. Congratulations! Your manuscript is now with our production department. 

Kind regards, 

on behalf of

Dr. Cinzia Ciccacci 

Academic Editor

PLOS ONE